# Estimating Households’ Expenditures on Disability in Africa: The Uses and Limitations of the Standard of Living Method

**DOI:** 10.3390/ijerph192316069

**Published:** 2022-12-01

**Authors:** Daniel Mont, Zachary Morris, Mercoledi Nasiir, Nanette Goodman

**Affiliations:** 1Center for Inclusive Policy, Washington, DC 20005, USA; 2School of Social Welfare, State University of New York-Stony Brook, Suffolk County, NY 11794, USA; 3Prospera, Jakarta 12190, Indonesia; 4Syracuse University College of Law, Burton Blatt Institute, Syracuse University, Syracuse, NY 13244, USA

**Keywords:** disability, Africa, extra costs, standard of living

## Abstract

People with disabilities face extra costs of living to participate in the social and economic lives of their communities on an equal basis with people without disabilities. If these extra costs are not accounted for, then their economic wellbeing will be overestimated. The Standard of Living (SOL) method is a way of generating these estimates and is thus useful for determining the economic impact of those costs in the current environment. However, previous studies have used different indicators for disability and different measures of the standard of living, so it is hard to compare estimates across different countries. This study applies a consistent set of indicators across seven African countries to produce comparable estimates. Our estimates of the extra costs of living in these lower-income countries are much lower than the results produced for higher-income countries in prior work. We argue that this finding highlights the limitations of the SOL method as a useful source of information for developing inclusive systems of social protection in lower-income countries because it captures what households spend but not what the person with a disability needs to fully participate in the social and economic lives of their community. In lower-income countries, people with disabilities are likely to have fewer opportunities to spend on needed items thus resulting in substantial unmet need for disability-related goods and services. Failing to account for these unmet needs can lead to inadequate systems of social protection if they are based solely on SOL estimates.

## 1. Introduction

Disability is correlated with poverty around the world [1], but that relationship is understated if the measure used is simply based on household income or consumption. When using multi-dimensional measures of poverty that relationship is stronger [2,3], especially when those indices account for the kinds of deprivations which are particularly associated with disability, such as social connections and autonomy [4,5]. However, another factor often left out of the analysis is the extra costs associated with living with disability.

People with disabilities face extra costs of living to participate in the social and economic lives of their communities on an equal basis with people without disabilities [6]. These costs include both those specific to people with disabilities, such as assistive devices and personal assistance, but also increased spending on mainstream needs like transportation and medical care. Thus, to have the same level of economic wellbeing as a household without a person with a disability, a household with a person with a disability must have income or consumption levels that are greater by the amount of those extra costs. A household with a member with a disability who cannot to meet all their basic needs at a particular level of income, may have been able to do so if they did not have a member with a disability who requires extra goods and services.

A household with a person with a disability whose consumption is at the poverty line has one of two options—either live below the poverty line and cover some, if not all, of the extra costs needed by their household member, or live at the poverty line (in terms of household items) but deny their household member with a disability those goods and services they need to live full lives. In either case, the person with a disability living in a household at the poverty line is, in essence, living below it. Additionally, even for families above the poverty line, those additional costs are likely to affect economic wellbeing.

When examining the impact of extra costs on the lives of people with disabilities, it is important to keep in mind the distinction between the extra costs needed for equal participation and the extra expenditures that are made by households with people with disabilities [7]. Household’s may not be able spend the full amount needed for equal participation for several reasons.

Families may not be able to afford the needed goods and servicesThe goods and services may not be available, especially in low income countries and in rural or remote areasPeople may be unaware of goods and services that can help them overcome barriers to participationDiscrimination can occur within a household, depriving people with disabilities of things they need

Most of the estimates of the extra costs of disability are estimates of extra expenditures, not the extra costs of goods and services required of equal participation. The estimates are useful in looking at the economic impact of disability on people’s lives, but not for determining people’s needs. Still, those estimates show a significant impact. When adjusting poverty lines by estimated extra expenditures on the needs of people with disabilities, the poverty rate in Cambodia rises from 18% to 34%, from 17.6% to 23% in Vietnam, 21.1% to 30.8% in Bosnia and Herzegovina, 32% to 42% in Mongolia and 38.5% to 52.9% in Ghana [8,9,10]. In the United States, adjusting the standard poverty line by disability-related expenditures would raise the estimated poverty rate of persons with disabilities from 24% to 35% and leave 85% of persons with disabilities in the US living below four times the poverty rate [11].

Most of the estimates of extra expenditures associated with disability rely on a methodology known as the Standard of Living Method (SOL), developed by Zaidi and Burchardt [12]. After briefly explaining this methodology, this paper addresses two challenges to this approach.

First, the SOL method relies on establishing a measure of the standard of living, and it is not clear how sensitive the results are to the chosen measure, and to what extent comparisons across countries are due to actual differences in extra expenditures or differences in how SOL is defined. This is not an indictment of the method, only a caution in comparing results that use different indicators for the standard of living—and comparing results between studies that use different indicators for identifying who has a disability.

Second, studies in South Africa and New Zealand, two very different countries, suggest that the gap between actual expenditures and the costs needed for full participation may be high [13,14]. In both cases, the estimated expenditures required for equal participation are very substantial and often significantly in excess many people’s incomes. As a result, it is not clear whether the SOL method, which does measure the current economic impact of disability expenditures on households, yields estimates that are appropriate for informing the design and budgeting for social protection policies and other schemes aimed at promoting full participation in LMIC.

Other limitations of the SOL are also discussed, for example that it only provides average expenditures, even though the South Africa and New Zealand studies show that these expenditures can vary widely across individuals, and the SOL does not provide information on what is being purchased. These two aspects of the SOL measure also limit its usefulness in designing programs to efficiently and adequately address these costs. Nevertheless, the SOL measure does provide information on the current average economic impact on households with members with disabilities, and sheds light on the inadequacy of poverty measures that do not take this into account. Therefore, it is useful in adjusting measures of disability poverty gaps. Moreover, if the sample size is large enough, SOL can be used to examine the difference in disability expenditures according to personal or household characteristics.

## 2. Standard of Living Method

The basic idea behind the SOL approach is that two families with the same income who are similar in a variety of other ways (e.g., household size, where they live, etc.) are expected to have the same level of wellbeing defined in this study as an asset index. If one of those households has a member with a disability, then any gap in wellbeing is assumed to result from the increased expenditures associated with the needs of the person with disability. In the absence of disability, those expenditures would be used to build up assets but for households with a disability, the expenditures are used to cover disability-related needs.

It is important to note that the SOL method does not address the indirect costs of disability, namely the foregone income of people with disabilities or household members. Those indirect costs result from barriers to employment or the need for household members to forgo paid work in order to provide support. The SOL method is only looking at the direct costs, that is extra expenditures on both disability specific and general items. Comparisons are being made between households that have the same level of income.

The approach developed by Zaidi and Burchardt is shown below in Figure 1. The higher line represents the relationship between income and standard of living for households without members with disabilities, the lower line is for those without members with disabilities. As income increases, the standard of living increases at the same rate for both types of households, but the line for households with members with disabilities is lower by the amount of those extra costs, which are assumed to be fixed. A household with a member with a disability must have an income of “*I2*” to have the same level of wellbeing as a household without a member with a disability with income “*I1*”. The line segment “*AB*” represents the extra cost of disability.

Zaidi and Burchardt formulate the standard of living approach as.
S = αY + βD + γX + k (1)
where S is an indicator of the standard of living, Y is household income, D is the presence of a household member with a disability, and X are other household characteristics. The parameter β is the impact of disability on the standard of living. Zaidi and Burchardt interpret k as the minimum level of standard of living a household needs to survive. The extra cost of disability, E, is given by
E = dY/dD = −β/α (2)

The distance between the lines is CB which is equal to β. The slope of the line is CB/AB equal to α. Thus β/α is CB/(CB/AB) which equals AB, or I2-I1, which is the extra cost of disability.

A common measure of the standard of living in the literature is wealth, represented by an asset index, but other indices could be used. For example, studies have used standard of living indicators based on self-rated financial satisfaction [15,16], the ability to afford different desired goods and services [17,18,19], or subjective assessment of the ability to make ends meet [20,21].

Because of the different definitions of the standard of living, it is unclear how comparable estimates are across countries. Even among studies that have used an asset index, the index has been constructed in different ways. It could be simply an index of the number of assets owned, an index constructed through principal component analysis (PCA), or a polychoric PCA [7,22]. The first of these three methods does not account for the correlation between the ownership of various assets, the second, which identifies a latent underlying variable of assets, does account for this, but the third extends that latent analysis to account for the impact of not owning an asset. For example, if nearly everyone has a particular asset, then having it does not add much information to an index, either based on the number of assets or even a PCA. However, with a polychoric PCA, the fact that one does not have an asset that is owned by most people can contribute to a signal that the person is poor [23].

## 3. Methodology

The goal of this study was to compare SOL estimates across countries in as comparable a manner as possible. To control for differences in estimates that might come from differences in how SOL is implemented, this study uses countries having data that allow for the similar construction of an asset index, the same covariates, and the same definition of disability. An effort was also made to control for differences that may result because of differences in how people with disabilities are identified across studies.

Survey questions used for identifying people with disabilities can vary significantly, and some types of questions have been shown to produce poor data [24]. The countries selected for this study were chosen because they all use the Washington Group Questions on disability, which have been widely adopted and recommended by many development agencies and international organizations and because they have surveys with data that can be used to construct an asset index in a similar manner. We also focused on lower income countries in one region, Sub-Saharan Africa, to make the results more comparable [25].

The countries with surveys meeting the above criteria were Ethiopia, Tanzania, Liberia, Nigeria, Namibia, Zimbabwe, and Malawi. Descriptive statistics for these countries’ data can be found in Table 1. A description of the survey design, number of observations, response rate, and links to more information about the surveys can be found in Table A1.

In all these surveys, a household was considered to have a member with a disability if at least one household member answered that they had “a lot of difficulty” or “cannot do” to at least one of the six activities in the Washington Group Short Set of Questions. These questions address seeing, hearing, walking or climbing steps, remembering or concentrating, understanding or being understood by others, and self-care (For the exact questions, and documentation on the use and testing of these questions see www.washingtongroup-disability.com (accessed on 29 November 2022)).

The other co-variates used in estimating Equation (1) were the log of consumption, age, age-squared, education of household head, household size, whether the household had health insurance, and whether they lived in an urban area. Regional dummies were also used. Consumption was used, and not income, because that is standard in low income countries where income is more difficult to measure and some households, particularly poor ones, may produce some of what they consume.

The same methodology was used to construct the asset index based on a comparable set of assets across countries, using the *tetrachoric* PCA command in Stata [26]. Tetrachoric PCA is a kind of polychoric PCA method that is used for estimating the principal component scores for binary variables. The first principal component scores for each country are included in the Appendix A. The mean value of the latent asset index for each country is in Table 1.

## 4. Results

The regression results for all seven countries are shown in Table 2, which also lists the names and years of the surveys used. In terms of the covariates other than disability, consumption and education were correlated with higher levels of assets across all the countries, as was household size, except for Namibia where it had a small negative, but statistically significant effect. Having health insurance also was associated with more assets in all countries, except for Ethiopia where there was no statistically significant correlation. Initially an additional year of age is correlated with more assets, but eventually, because of the negative coefficient on age-squared, it starts being associated with decreasing assets. These results are generally as expected.

As for disability, the main variable of interest, it is always negatively associated with the asset index, but is only statistically significant in three of the seven countries: Tanzania, Nigeria, and Zimbabwe. As described above, the estimates for the extra expenditures associated with disability are estimated by way of the ratio of the coefficient for disability and the coefficient for log consumption. Thus, for Ethiopia, we estimate that households including a member with a disability require 6% (−0.012/0.212) more income to maintain their living standards relative to a comparable household without a member with a disability. These estimates are lower than is often seen in higher income countries, as discussed below. It ranges from 4% in Zimbabwe to 10% in Tanzania of household consumption and is not statistically significant in Liberia and Namibia.

## 5. Discussion

The estimates for the extra expenditures associated with disability contrast sharply with estimates in higher income countries, where the extra expenditures average out at about 43%, as show in Table 3. As stated earlier, this could be because the reasons for the gaps between what is spent and what is needed are much stronger in poorer countries. In fact, Table 4 shows that the two poorest countries analyzed in this study did not show significant increased expenditures. Namibia, however, is the exception. It is by far the country with the highest income among the countries studied and shows no association between disability and extra expenditures. Of course, it is still a relatively poor country. The average per capita GDP in the world in 2020 was $10,961, compared to $4179 in Namibia. However, these numbers also contrast with a study done in Ghana, a similar African country with GDP per capita equal to $2206 using a PCA asset index, though not far from another country with a similar level of income, Vietnam ($2656 per capita), at 11% [10,27].

The SOL is a valuable tool to estimate current average economic impact of the extra expenditures needed for people with disabilities to participate. However, it has significant limitations:

First, it does not account for the indirect costs of disability, that is foregone income, only the direct costs incurred by households including members with a disability. Not accounting for these direct costs is, nevertheless, important as it will overestimate the economic wellbeing of households with members with disabilities. Poverty will be underestimated among households with people with disabilities if the SOL method is not used to adjust poverty rates. However, given the wide range of needs—as demonstrated in the South African and New Zealand studies mentioned above—that average hides a great deal of variance. By using the average cost, we will often under or overstate the economic impact of disability on households. Some households have very high costs and some have very low costs. For example, one person may only need a walker, but another might need a personal assistant and a respirator. Moreover, these differences can very well be correlated with the type and degree of disability. Only looking at the average might unintentionally give the impression that a simple top-up of a cash transfer program can adequately address everyone’s needs.

Second, in low income countries where many people are highly income constrained and are more likely to lack knowledge of or access to goods and services needed for participation, disability-related expenditures may not be substantial. Unable to purchase needed items, people with disabilities in such low-resource environments are likely to go with substantial unmet needs for disability-related goods and services. If designers of social protection policies aimed at equalizing economic wellbeing between households with and without disabilities do not take this into account, they could conclude that a relatively minor top-up to a cash transfer for households with members with disabilities is sufficient. This could be incorrect if estimates of what people are currently spending are small, not because they do not need substantial goods and services but because they cannot afford them or they are not currently available. A cash top-up may—on average—equalize resources available for non-disability related expenditures, but it would be woefully insufficient for providing the resources necessary to purchase the goods and services required for full participation in economic and social life.

The large variance in how costs are incurred could call for a set of programs, for example programs targeted at large expenses, like personal assistance or assistive devices, as well as concessions for areas where costs are higher for people with disabilities (such as transportation costs) as well as cash benefits to cover various idiosyncratic costs.

While the SOL can highlight that disability is leading to extra expenditures and so draw attention to these issues, it is important that policymakers go beyond the SOL to measure the types of goods and services people with different types and degrees of disability require so they can match the structure of government programs to the structure of how those costs are incurred. Work to make such estimates, as described in [7] and drawing upon the methodologies in New Zealand and South Africa described elsewhere [13,14], are currently underway in Georgia, Peru, and Tamil Nadu, and will hopefully be available soon. Further research on estimating the types of goods and services need

## 6. Conclusions

The SOL method is a powerful approach using widely available data to estimate the economic impact of direct disability costs in the current environment. When making cross-country comparisons, it is important to be cognizant of differences in how both the standard of living and disability are defined. This study provides a cross country comparison in one region that uses common indicators.

The SOL method, however, is not well suited for the design of social protection programs aimed at inclusion. Especially in low income countries, SOL estimates can mistakenly be used to draw the conclusion that only minor benefits are needed because in their current environment, for various reasons, not many expenditures occurring. The design of such program requires information on the types of goods and services that people with disabilities would need to participate on an equal basis with their non-disabled peers.

## Figures and Tables

**Figure 1 ijerph-19-16069-f001:**
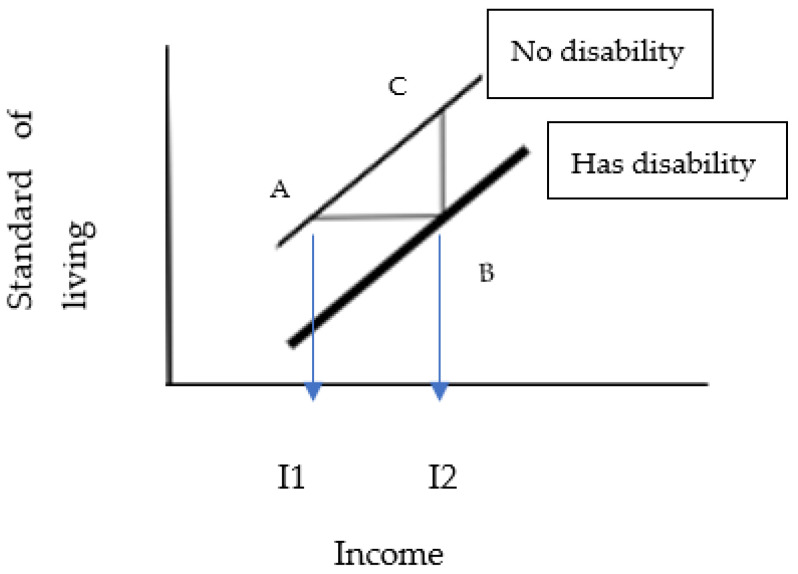
Relation Between Standard of Living and Income.

**Table 1 ijerph-19-16069-t001:** Descriptive statistics by country and according to households with and without a person with a disability (unweighted).

	Disability (Mean)	SD	No Disability (Mean)	SD
Ethiopia				
Consumption (log)	7.01	0.73	7.23	0.79
Age of household head	54.87	16.97	41.75	14.47
Education of household head (years)	8.07	4.22	8.84	4.54
Urban	0.50	0.50	0.59	0.49
Household size	5.41	2.35	4.34	2.23
Health insurance	0.18	0.38	0.17	0.38
Asset Index	0.77	1.28	1.04	1.28
Observations	633		6137	
Tanzania				
Consumption (log)	11.33	0.90	11.53	0.94
Age of household head	57.30	16.34	42.57	14.97
Education of household head (years)	0.72	0.45	0.87	0.34
Urban	1.38	0.49	1.44	0.50
Household size	3.42	1.18	2.92	1.06
Health insurance	0.08	0.27	0.07	0.26
Asset Index	1.75	1.02	1.91	1.08
Observations	234		950	
Liberia				
Consumption (log)	7.65	1.02	7.72	0.95
Age of household head	47.81	14.71	41.73	20.67
Education of household head (years)	0.14	0.19	0.20	0.26
Urban	0.28	0.45	0.34	0.47
Household size	3.08	0.99	2.76	0.97
Health insurance	0.02	0.14	0.02	0.13
Asset Index	0.92	0.67	0.92	0.68
Observations	814		7536	
Nigeria				
Consumption (log)	11.93	0.63	12.11	0.68
Age of household head	55.33	17.70	47.88	15.26
Education of household head (years)	0.73	0.44	0.82	0.39
Urban	1.76	0.43	1.68	0.46
Household size	6.08	3.78	5.08	3.20
Health insurance	0.02	0.14	0.04	0.20
Asset Index	1.00	0.99	1.24	1.07
Observations	2993		19130	
Namibia				
Consumption (log)	10.88	0.98	11.00	1.00
Age of household head	59.28	18.20	44.87	16.21
Education of household head (years)	0.71	0.46	0.84	0.37
Urban	1.67	0.47	1.53	0.50
Household size	5.75	3.54	3.90	2.73
Health insurance	0.02	0.15	0.09	0.29
Asset Index	1.11	1.14	1.38	1.23
Observations	1222		8868	

Note: SD = Standard deviation. Disability defined as moderate-severe.

**Table 2 ijerph-19-16069-t002:** Regression Results for SOL.

Dependent Variable: Latent Asset Index	Ethiopia	Tanzania	Liberia	Nigeria	Namibia	Zimbabwe	Malawi
**Survey**	Ethiopia Socioeconomic Survey 2018/2019	The National Panel Survey 2019/2020	Household income and expenditure survey (2016)	Nigeria Living Standards Survey 2018/2019	Namibia Household Income and Expenditure Survey, 2015/2016	Poverty, Income, Consumption and Expenditure Survey Questionnaire 2017	Integrated Household Survey and Integrated Household Panel Survey 2019
**Extra expenditure estimate**	**6%**	**10%**	**Not significant**	**8%**	**Not significant**	**4%**	**Not significant**
Person with moderate-severe disability in household	−0.012	−0.056 **	−0.003	−0.053 ***	−0.007	−0.014 ***	−0.009
Consumption (log)	0.212 ***	0.578 ***	0.261 ***	0.692 ***	0.555 ***	0.382 ***	0.337 ***
Age	0.305 ***	0.008	0.122 ***	0.012 ***	0.018 ***	0.132 ***	0.435 ***
Age squared	−0.290 ***	0.048	−0.094 ***	−0.000 ***	−0.000 ***	−0.126 ***	−0.233 ***
Education of household head	0.237 ***	0.122 ***	0.172 ***	0.367 ***	0.308 ***	0.145 ***	0.337 ***
Urban	−0.598 ***	0.157	0.267 ***	−0.763 ***	−0.711 ***	0.431 ***	0.248 ***
Regional dummies	Included	Included	Included	Included	Included	Included	Included
Household size	0.035 ***	0.219 ***	0.253 ***	0.072 ***	−0.018 ***	0.215 ***	0.171 ***
Health insurance	−0.005	0.048 *	0.084 ***	0.679 ***	0.753 ***	0.031 ***	0.107 ***
Observations	5846	1080	8287	20,750	10,090	26,431	11,404

* is significant at the 90% confidence level; ** is significant at the 95% confidence level; *** is significant at the 99% confidence level.

**Table 3 ijerph-19-16069-t003:** Extra expenditures as a proportion of household consumption in high income countries.

Country	Percent	Country	Percent
USA	29	Ireland	41
UK	51	Iceland	77
Switzerland	54	Hungary	16
Spain	41	Germany	35
Slovenia	52	Greece	32
Slovakia	25	France	29
Romania	40	Finland	78
Portugal	38	Estonia	27
Poland	16	Denmark	56
Luxembourg	36	Czech Republic	36
Norway	89	Cyprus	17
Netherlands	63	Croatia	27
Malta	53	Bulgaria	21
Lithuania	30	Belgium	36
Latvia	37	Austria	54
Italy	45	Australia	50
AVERAGE		43	

Sources: Antón, J. I. et al. (2016). An analysis of the cost of disability across Europe using the standard of living approach. *SERIEs*, *7*(3), 281–306 [20]. Ozdamar et al. (2020) [28]; Touchet, A., & Morciano, M. (2019). *The Disability Price Tag 2019*. Technical Report [29]; Morris, Z.A. et al. (2022). The extra costs associated with living with a disability in the United States. *Journal of Disability Policy Studies*, 10442073211043521 [11]; Vu, B. et al. (2020). The costs of disability in Australia: a hybrid panel-data examination. *Health Economics Review*, *10*(1), 1–10 [16]; Palmer, Carraro, L. & Cumpa, M.C. (2014). Accounting for different needs when identifying the poor and targeting social assistance. Paper prepared for the IARIW 33rd General Conference, Rotterdam, 24–30 [30]; Amin, R.M., & Adros, N.S.M. (2019). The Extra Costs of Having a Disability: The Case of IIUM. *Intellectual Discourse*, *27*(SI# 2), 829–854 [31].

**Table 4 ijerph-19-16069-t004:** Estimates of Extra Expenditures by Per Capita GDP.

	Per Capita GDP in 2020 ^a^	Estimate of Extra Expenditures
Liberia	632.9	Not significant
Malawi	636.8	Not significant
Ethiopia	963.3	6 percent
Tanzania	1076.5	10 percent
Zimbabwe	1214.5	4 percent
Nigeria	2097.1	8 percent
Namibia	4179.3	Not significant

^a^ Source: https://data.worldbank.org/indicator/NY.GDP.PCAP.CD (accessed on 29 November 2022).

## Data Availability

Namibia Household Income and Expenditure Survey, 2015/2016: https://nsa.org.na/microdata1/index.php/catalog/28 (accessed on 29 November 2022); Nigeria Living Standards Survey 2018–2019: https://microdata.worldbank.org/index.php/catalog/3827 (accessed on 29 November 2022); Ethiopia Socioeconomic Survey 2018/2019: https://microdata.worldbank.org/index.php/catalog/3823 (accessed on 29 November 2022)); Tanzania National Panel Survey 2019/2020: https://microdata.worldbank.org/index.php/catalog/3885 (accessed on 29 November 2022)); Liberia Household Income and Expenditure Survey 2016: https://ghdx.healthdata.org/record/liberia-household-income-and-expenditure-survey-2016-2017 (accessed on 29 November 2022)); Zimbabwe2017 Poverty Income Consumption Survey: https://catalog.ihsn.org/catalog/9250 (accessed on 29 November 2022)); Malawi Fifth Integrated Household Survey 2019–2020 Malawi, 2019–2020: https://microdata.worldbank.org/index.php/catalog/3818 (accessed on 29 November 2022)).

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
