# Peer review of "Estimating Households’ Expenditures on Disability in Africa: The Uses and Limitations of the Standard of Living Method"

_ijerph, 2022, doi:10.3390/ijerph192316069_

Round 1

Reviewer 1 Report

Statistics have never been my strong suit. Nevertheless, the findings the authors present are critically important. The article examines an underutilized method of comparison on the basis not of income but on services and the materials necessary for households inclusive of person(s) with disability require more than a stipend can supply.

It would be helpful to readers to have explicit examples of the difference that goods and services provide to households inclusive of disability.

Author Response

Thank you for your comments and your agreement that this is an important area.

We added some language to better elucidate the usefulness of particular goods and services and expanded the discussion section and added a conclusion section to more fully present our argument.

Reviewer 2 Report

It is my pleasure to review the paper “Estimating Households’ Expenditures on Disability in Africa: 1 The uses and limitations of the Standard of Living Method” to be considered for publication at Int. J. Environ. Res. Public Health. My comments follow the structure of the paper.

Abstract.

The paper states that it “highlights the limitations of the SOL method” as such I would have expected some methodological contributions on how this study was able to refine the SOL method. However, such insights are currently lacking. Instead there is a more common-sense statement “lower-income countries, people with disabilities are likely to have fewer opportunities to spend on needed items thus resulting in substantial unmet need for disability-related goods and services”. Again I would have expected the paper to elaborate how the method employed by authors is superior to SOL.

Introduction.

The phrases which seemingly are central to the argument of the paper are not clear :”Most of the estimates of the extra costs of disability are estimates of extra expenditures, not the extra costs of goods and services required of equal participation. The estimates are useful in looking at the economic impact of disability on people’s lives, but not for determining people’s needs.”. It seems the authors try to argue on terminology rather than content basis for differences in their study. Perhaps an example would further help.

I like to cite Zaidi and Burchardt (2003) “The standard of living approach for measuring extra costs of disability draws on the subjective approach to equivalisation by concentrating on the relationship between standard of living and income, and on the consumption-patterns approach by using objective data on incomes and consumption. At the same time, the standard of living approach does not suffer from many of the drawbacks associated with other approaches.

The debate between conditional and unconditional measures is irrelevant, since, first, no-one chooses to be disabled; and second, the standard of living approach is restricted to measuring the extra costs of living: it does not attempt to make an overall welfare comparison (in which case an assessment of the direct utility or disutility of being disabled would be necessary). In contrast to the subjective approach and minimum budget approach, for the standard of living method neither individuals nor experts are required to make judgements about hypothetical levels of consumption with and without disability. Instead, the differences are deduced from observations of the relationship between standards of living and income.”

The Zaidi and Burchard clearly state they draw on relationship between standard of living and income and incomes and consumption. The sensitivity of these measure are an empirical question. The point to make the authors propose a contribution claiming to contest SOL on measures which were clearly not covered by SOL. As such it would require authors to discuss conceptually how this paper is positioned.

3.Methodology

This section is lacking further details: How were the surveys distributed, how representative are the samples, how common method bias addressed? Unfortunately none of these issues have been addressed.

4. Results

Adding all observations into one regression without testing if there is a significant difference between the samples is highly problematic. Authors made the assumption that there are no country level difference which could drive regression results.

5. Discussion

The discussion reads very superficial and needs further reference to existing literature. It is not clear how these findings are better or enhance SOL measure.

The section conclusion is missing.

Reference

More citations of Int. J. Environ. Res. Public Health could help to enhance the relevance of this study to the journal.

Author Response

Thank you for your comments. They are very helpful because it shows we weren't clear enough in describing the purpose of the paper.

We are not suggesting methodological improvements for what SOL estimates. We are trying to show how those results should be interpreted and used.

What SOL does , as Reviewer 2 states, is measure the current average level of expenditures associated with disability and thus helps see the current economic impact of disability on household's economic wellbeing. What it does not do is estimate what level of expenditures is needed to ensure full participation of people with disabilities.  In a very rich country these two thing might not be that different, given social protection programs are available, fewer barriers exist, people are aware of the goods and services they may need, and they are generally available.

However, in lower income countries many of these things are not true. Therefore, although people may be spending $100 on disability costs, the people with disabilities in their family may be facing exclusion from school, work, family life, civic life, etc. because the family would need to spend $500 in order to enable the person with a disability to participate on an equal basis with non-disabled people.

Our concern is that people will mis-use the SOL estimates. They will say -- oh, it seems families spend an extra $100 so what we need to do in our social protection programs is give a cash benefit of $100 and we have offset disability costs. True, (on average) they may offset the current economic impact on households with people with disabilities, but that will not provide the support needed for full participation -- which is the goal of the Convention on the Rights of Persons with Disabilities that nearly all countries have ratified.

We chose low income countries to analyze because we suspected that in these countries where people have serious budget constraints and less knowledge and access to the goods and services they need for full participation (e.g., assistive technology, rehabilitative services, etc.) that the extra expenditures they are making could be quite low. That means that although they are not currently  incurring a big direct cost on their households the level of exclusion may be high. 

The low level of costs uncovered in this paper is evidence to us that the SOL is not a good measure for use in designing social protection programs aimed at promoting inclusion -- even if it is a good measure for estimating the average economic impact within the current environment. This is especially true for low income countries. As the countries develop, income rises, and availability and knowledge of goods and services required for participation rises, we expect the costs measured by the SOL to increase.  That is the main point of our paper.

A second point is that since different studies use different measures of wellbeing, and/or different definitions of disability, the different cost estimates we see between countries may result not from real differences in costs, but differences in disability and wellbeing indicators. Almost all studies using SOL have focused on one country, and the studies looking at multiple countries have with only one exception we are aware of been for high income countries. Therefore, we chose six low income countries (from the same region to minimize differences) that used the same definition of disability and for which we could construct similar asset indices as measures of the standard of living.

In this paper we are not proposing any change in the SOL measure for what we consider its useful purpose -- measuring the economic impact on households with people with disabilities in the current environment.

We believe, however, for the design of social protection systems, we need to estimate the goods and services required for full participation.

In the revised paper we have tried to make this clearer, in part with an expanded discussion section and a conclusion.

We also better describe the data sets we've used, as requested.

Reviewer 3 Report

This is a well written and important paper. I only have a few areas where small revisions are suggested:

1) Pg 1, sentence beginning line 40: would be good to revise/clarify the sentence to reflect that just having the additional income is not always enough to meet extra costs (discussed later but helpful to be consistent throughout).

2) Pg 2, line 44: could clarify that it's not just people living below the poverty line that deal with extra costs.

3) Pg 2, paragraph beginning with line 88: In theory, SOL results could be disaggregated by individual characteristics if there is a sufficient sample size. 

3) Page 4, paragraph beginning line 149: in several places it says (cite) - is this missing a reference? 

4) Methods: what was the selection process for picking surveys? Countries with Washington Group Questions, but were you initially looking globally or only in Africa?

5) Discussion - Tables 3 and 4 referenced in text but I don't see the corresponding tables

6) The end of the conclusion could be strengthened as currently ends a bit abruptly. Discussion on areas for future research, implications for policy? 

Author Response

Thank you for your comments. We feel they will improve the paper.  In response to them:

1) We added the following sentence at line 40 to emphasize this point: 

A household with a member with a disability who cannot to meet all their basic needs at a particular level of income, may have been able to do so if they did not have a member with a disability who requires extra goods and services.

.

2)At what was line 44 we have added the sentence:

And even for families above the poverty line, those additional costs are likely to affect economic wellbeing.

3)  We added the sentence:

Moreover, if the sample size is large enough, SOL can be used to examine the difference in disability expenditures according to personal or household characteristics.

3) (there were two points labeled 3). We put in the citations. Thanks for catching that.

4) We edited the paragraph to describe why we chose those countries. It's copied below. We also added text describing the data sets in more detail.

The countries selected for this study were chosen because they all use the Washington Group Questions on disability, which have been widely adopted and recommended by many development agencies and international organizations and because they have surveys with data that can be used to construct an asset index in a similar manner. We also focused on lower income countries in one region, Sub-Saharan Africa, to make the results more comparable[25].

5) We put in table 3 and 4. That was a pretty bad oversight on our part!

6) We expanded the discussion section and added a conclusion section that we think explains things more fully and points the way to future research.
